# Small but Mighty—Exosomes, Novel Intercellular Messengers in Neurodegeneration

**DOI:** 10.3390/biology11030413

**Published:** 2022-03-08

**Authors:** Meena Kumari, Antje Anji

**Affiliations:** 1Department of Anatomy and Physiology, College of Veterinary Medicine, Kansas State University, Manhattan, KS 66506, USA; 2Bureau of Epidemiology and Public Health Informatics, Kansas Department of Health and Environment, Topeka, KS 66612, USA; antjeanji@gmail.com

**Keywords:** exosomes, neurons, glial cells, central nervous system (CNS), neurodegenerative diseases, Alzheimer’s disease, Parkinson’s disease, amyotrophic lateral sclerosis

## Abstract

**Simple Summary:**

Exosomes are biological nanoparticles recently recognized as intercellular messengers. They contain a cargo of lipids, proteins, and RNA. They can transfer their content to not only cells in the vicinity but also to cells at a distance. This unique ability empowers them to modulate the physiology of recipient cells. In brain, exosomes play a role in neurodegenerative diseases such as Alzheimer’s disease and Parkinson’s disease and amyotrophic lateral sclerosis.

**Abstract:**

Exosomes of endosomal origin are one class of extracellular vesicles that are important in intercellular communication. Exosomes are released by all cells in our body and their cargo consisting of lipids, proteins and nucleic acids has a footprint reflective of their parental origin. The exosomal cargo has the power to modulate the physiology of recipient cells in the vicinity of the releasing cells or cells at a distance. Harnessing the potential of exosomes relies upon the purity of exosome preparation. Hence, many methods for isolation have been developed and we provide a succinct summary of several methods. In spite of the seclusion imposed by the blood–brain barrier, cells in the CNS are not immune from exosomal intrusive influences. Both neurons and glia release exosomes, often in an activity-dependent manner. A brief description of exosomes released by different cells in the brain and their role in maintaining CNS homeostasis is provided. The hallmark of several neurodegenerative diseases is the accumulation of protein aggregates. Recent studies implicate exosomes’ intercellular communicator role in the spread of misfolded proteins aiding the propagation of pathology. In this review, we discuss the potential contributions made by exosomes in progression of Alzheimer’s disease, Parkinson’s disease, and amyotrophic lateral sclerosis. Understanding contributions made by exosomes in pathogenesis of neurodegeneration opens the field for employing exosomes as therapeutic agents for drug delivery to brain since exosomes do cross the blood–brain barrier.

## 1. Introduction and Historical Perspective

Intercellular communication is of great interest for embryonic development, regeneration/replacement of epithelia, gametogenesis, and maintenance of homeostasis. Several mechanisms of cell–cell communication such as endocytosis/exocytosis and engulfing protruding cytoplasmic blebs were identified using ultrastructural and biochemical studies [1,2,3]. Another player, extracellular vesicles, was identified but their role in intercellular communication was not appreciated immediately [4]. Extracellular vesicles were discovered serendipitously during an investigation on release of plasma membrane-associated enzymes, ecto-ATPase and ecto-5′-nucleotidase into cell culture medium. In 1981, Trams and coworkers cultured normal and neoplastic cell lines of mouse and human origin in serum-free medium because serum was known to have catalytic activity of ATPase and 5′-nucleotidase. After removal of floating cells and debris, the conditioned medium was centrifuged at a speed of 310,000× *g* for 90 min in a Spinco Ti-70 rotor and the pellets were examined under the electron microscope. Two populations of extracellular membrane-enveloped vesicles were discovered, one with an average diameter between 500 and 1000 nm and a second, smaller population with a diameter of ~40 nm. These vesicles, initially described as “exfoliated vesicles” by authors, contained both 5′-nucleotidase and ATPase enzymatic activity. In addition, the exfoliated vesicles contained RNA and phospholipids that were different from parent cells and parent plasma membrane, respectively. Incubation of exfoliated vesicles from C6 glioma cells with ^32^P-prelabeled neuroblastoma cells stimulated de-phosphorylation of neuroblastoma proteins with concomitant release of free radiolabeled phosphorus (^32^P). These data established that the exfoliated vesicles were not merely a carrier of cellular waste (enzymes) but had a physiological function. Tram and colleagues coined the term exosomes to describe the “exfoliated vesicles” [5] (see review article by [6] for extracellular vesicle history prior to 1981). Some years later, two research groups independently studied the fate of the transferrin receptor in reticulocytes by electron microscopy [7,8,9]. Maturing reticulocytes do not require transferrin so they lose transferrin receptors. Both groups observed that cultured reticulocytes released small vesicles of ~50 nm in diameter into the extracellular space. These small vesicles contained un-degraded transferrin receptors. In 1987, Johnston and co-workers used the term exosomes to describe small vesicles containing transferrin receptors [10]. Following these initial years, the potential of exosomes was realized as novel intercellular messengers that can modulate the physiology of target cells. Interestingly, genesis of exosomes is initiated with endocytosis (a process identified in 1883; reviewed by [3]) in cells and exosome release to intracellular space uses the process of exocytosis. 

Since their initial discovery, we now know that exosomes are a subtype of extracellular vesicles of endocytic origin with a size range of 40–160 nm in diameter [11]. Tracer studies demonstrated that macromolecules are endocytosed in bristle-coated (i.e., clathrin-coated) invaginations or pits on the cell surface. These invaginations eventually separate from the plasma membrane and are detected in the cytoplasm. After losing bristles (i.e., clathrin), endocytosed coated vesicles become smooth and fuse with early endosomes for either recycling rapidly back to the cell surface (recycling endosomes in Figure 1) [12] or undergo maturation to form late endosomes. Early and late endosomes act as “sorting hubs” to appropriately direct their content: (i) for retrograde transport to the *trans*-Golgi network (i.e., endosome-to-TGN retrieval) [13,14]; (ii) to lysosomes for degradation of macromolecules to simple products for reuse within the cell [15,16]; and/or (iii) for recycling to the cell surface through multivesicular bodies [17]. During maturation of early endosomes into late endosomes, the internal endosome membrane undergoes invagination, resulting in the formation of intraluminal vesicles (now designated as exosomes). Specific macromolecules such as cytosolic proteins, nucleic acids, and lipids are sorted into developing exosomes. Late endosomes packed with exosomes are identified as multivesicular bodies that migrate towards the plasma membrane. Upon fusion with the plasma membrane, multivesicular bodies release exosomes into the extracellular space through the process of exocytosis (Figure 1) [18,19]. A large subset of proteins that are sequentially recruited play a decisive role from the process of endocytosis to early endosome, endosome maturation to late endosomes, and their subsequent intracellular fate (reviewed by [17,18,20]). The importance of these proteins is highlighted by their loss and/or mutation(s) that cause various diseases. For instance, Strumpellin and SWIP proteins are a part of the WASH complex that participates in actin polymerization, as well as endosome fission and sorting. Mutations in the *Strumpellin* gene causes hereditary spastic paraplegia, characterized by degeneration of motor neurons [21]. A single point mutation (Pro1019Arg) in the SWIP protein destabilizes the WASH complex. The destabilized WASH complex inhibits endolysosomal trafficking, causing clumping of endosomes. Defective endosomal traffic impacts cognition and motor neuron function, symptoms described as nonsyndromic mental retardation [22,23].

In the cytoplasm, exosomes originate from endosomes using the endosomal sorting complexes required for transport (ESCRT), also known as the ESCRT-dependent mechanism of exosome biogenesis (reviewed by [18,20]). Two groups independently reported lipid-mediated biogenesis of exosomes, an ESCRT-independent mechanism. Two lipids identified in these studies were bis(monoacylglycero)phosphate/lysobisphos phatidic acid (BMP/LBPA) and ceramide [29,30]. Rab31 is another ESCRT-independent mechanism for exosome biogenesis in the cytoplasm. Activation of Rab31 stimulates formation of intraluminal vesicles [31]. There is another mechanism that is currently under-discussed and under-explored. This mechanism is based upon ultrastructural observations made in the late 1950s to 1970s that showed cell nuclei can be a site for origin multivesicular bodies packed with small vesicles. Small vesicles described in these studies were perhaps exosomes but the term exosome was not coined at that time. Kilarski and Jasinski observed a process termed nuclear blebbing or nuclear extrusions in cells of the swim bladder of the perch fish, *Perca fluviatilis* L. (Figure 2a) [24] with content similar to nucleolar karyoplasm [32]. These authors noted nuclear blebbing can be stimulated by removal of gas from the swim bladder of the perch fish [24]. The process of nuclear blebbing appears to have merit as the site of origin of exosomes enriched with nucleic acids and/or ribonucleoproteins. Nuclear blebs were filled with small vesicles that arose from single in-foldings of the internal membrane of the nuclear blebs (Figure 2b), a process akin to the formation of intraluminal vesicles in endosomes. Nuclear blebs filled with ‘small vesicles’ eventually separated from nuclei to become part of the cytoplasm (Figure 2c). Kilarski and Jasinski referred to nuclear blebs as multivesicular bodies in their manuscript due to lack of a better terminology. Inadvertently, these authors might have been right as subsequent studies reported the presence of multivesicular bodies in nuclei of cells [33]. The nuclear blebs filled with small vesicles have morphology similar to multivesicular bodies originating from endosomes in the cytoplasm (Figure 1 and Figure 2). Because of the similarity of nuclear multivesicular bodies with viral nucleocapsid, researchers focused their attention on virology and the phenomenon of nuclear blebbing and its cellular function remained underexplored [34]. Irrespective of the mode of biogenesis, exosomes are composed of a lipid bilayer and contain macromolecular content consisting of lipids, proteins, and nucleic acids. Some of the macromolecules are common to exosomes originating from all cells and hence are classified as exosome markers and are used as a confirmatory tool to distinguish exosomes from other extracellular vesicles. Unique macromolecules present in exosomes often reflect their parental origin and hence are likely to be designated as biomarker(s).

All cells release exosomes into the extracellular space. From the extracellular space, exosomes find their way into various biological fluids and reach cells locally and at a distance. Cells cultured in vitro release their exosomes into culture media. Continuous release of exosomes into the extracellular space makes culture medium and biological fluids a rich source for the isolation of exosomes.

Exosomes contain a substantial amount of coding and non-coding RNAs, double-stranded genomic DNA fragments, and mitochondrial DNA. Trams and colleagues were the first to describe the presence of RNA in “exfoliated vesicles”, a mixture of exosomes and microvesicles. They determined that “exfoliated vesicles or exosomes” contain 5% RNA [5]. Valadi and colleagues were the first to systematically analyze RNAs packaged in exosomes. The exosomal RNAs included translatable mRNAs and microRNAs (miRs) [35]. MicroRNAs are a class of non-coding RNAs (ncRNAs). They are ~22 nucleotides in length and regulate gene expression at the post-transcriptional level by binding to 3′-untranslated regions of their target messenger RNAs [36]. Interestingly, some miRNAs enriched in exosomes are not detected in the parent cells suggesting a selective RNA sorting/packaging system during biogenesis of exosomes [35,37]. Unlike cells, exosomes contain little or no ribosomal RNA but do contain tRNA fragments [38]. Sorting of RNA into exosomes depends upon: (1) RNA-induced silencing complex (RISC)-associated protein, (2) RNA sequence motifs and guide proteins, (3) nucleotide additions at the 3′ end of the miRNAs, (4) cellular availability of miRNAs, and (5) ceramide [39,40]. We believe that nuclear blebbing is another cellular process that allows differential sorting of RNA, RNA–protein complexes, and genomic DNA into exosomes. There are several reasons for this viewpoint: (1) the content of ‘intraluminal vesicles’ was similar to nucleolar karyoplasm [32]. ‘vesicles’ in nuclear multivesicular bodies (MVB II) were enriched with RNA, acidic and basic proteins, and hence authors considered ‘intraluminal vesicles’ as the site of RNA synthesis in addition to nucleolus [41]. Proteins present in the vesicles may have been RNA-binding proteins and Dicer1 that were detected in exosomes [42] (unpublished data, Kumari Lab, Kansas State University, Manhattan, KS, USA). (2) Considering the nucleus as a site of exosome biogenesis can explain the presence of RNAs including precursor miRNAs (unpublished data, Kumari Lab) and double-stranded genomic DNA fragments ranging from 100 bp to >10 kb in exosomes [43,44,45,46,47]. Incidentally, genomic DNA was detected only in exosomes derived from cancer cell lines and not normal cells [47]. Cancer cells have abnormal chromatin due to chromatin rearrangement/translocation and perhaps ploidy [48] that can cause genomic DNA fragmentation in cancer cell nuclei. These DNA fragments can easily be incorporated into MVB type II (and exosomes) developing in the nuclei. (3) The mode of ‘small vesicle’ biogenesis in nuclear blebs described by Kilarski and Jasinski [24] is similar to the formation of intraluminal vesicles in late endosomes in the cytoplasm. We strongly believe that the presence of genomic DNA in exosomes is pathological and hence its presence in exosomes can be used as a diagnostic tool for the development of abnormal (or cancer) cells in the body. (4) The activity-dependent increase in the number of exosomes released by cells [49] is comparable to gas cells that showed increased nuclear blebbing upon removal of gas from the swim bladder of perch fish [24]. More recently, mitochondrial proteins and DNA were detected in plasma exosomes of some cancer patients and their presence is associated with aggressiveness of the cancer [50,51]. Mitochondrial breakdown is often observed in degenerating cells [52,53]. Mitochondrial cristae released from the degenerating mitochondria may be internalized by multivesicular bodies and are eventually packaged in exosomes. This notion is supported by ultrastructural images of human lymphoblastoid cells and lumen of acini of the baboon prostate that had ‘microvesicles’ (or exosomes) derived from the breakdown of the cristae of mitochondria [54]. Based upon the above discussion and supportive experimental evidence, we propose that the heterogeneous nature of exosomes originates from biogenesis of multivesicular bodies and associated exosomes at different subcellular locations and not entirely in the cytoplasm as per our current belief. Our proposal can explain the heterogeneity of exosomes as well as a mismatch between theoretical calculations and experimental data on the number of small RNAs or protein molecules contained within one exosome purified from serum [55]. Based upon theoretical calculations using average exosome size, Li and colleagues predicted each exosome to accommodate approximately 70–25,000 small RNA or protein molecules. However, their experimental data showed the presence of one RNA molecule per exosome purified from serum [55]. Our hypothesis necessitates further exploration on nuclear blebbing, nuclear multivesicular bodies and biogenesis of RNA/ribonucleoprotein-containing exosomes in nuclear multivesicular bodies. 

Lipids are an important part of exosome membranes and are distributed asymmetrically between the two exosome membrane leaflets. Lipid composition differs between parent cell membrane and exosome membrane. In general, lipids enriched in exosome membranes are sphingomyelin, ceramide, cholesterol and cholesterol-rich lipid raft microdomains, ganglioside GM3, disaturated lipids, and phosphatidylserine [56,57,58].

Exosomal proteins are comprised of those common to most exosomes as well as unique proteins specific to a certain parent cell. The protein content of all exosomes is generally enriched for targeting and fusion proteins such as tetraspanins, intergrins, chaperone proteins, e.g., heat shock proteins (Hsp60, Hsp70, and Hsp90), the small HSP membrane trafficking proteins, Rab proteins, ARF GTPases, annexins, and proteins involved in the formation of multivesicular bodies such as ALIX and TSG101 [59]. More recent studies suggest the presence of RNA-binding proteins that form RNA–protein complexes in exosomes [38]. A number of cellular processes that ensure uploading of proteins into exosomes are: (1) ESCRT mechanisms for proteins with certain post-translational modifications such as ubiquitination, Wnt-stimulated arginine methylation, oxidation, phosphorylation, glycosylation, and citrullination; (2) chaperone-mediated autophagy (CMA) that relies on interaction between heat shock cognate 71 kDa protein (HSC70) and proteins with a KFERQ motif. After binding of HSC70 to proteins with a KFERQ motif, proteins are targeted to multivesicular bodies. Chaperone-mediated autophagy normally operates for degradation of proteins in lysosomes. More recent data suggest that CMA plays a role in exosome release and exosome-mediated dissemination of protein aggregates in some neurodegenerative diseases [60,61,62,63].

There has been an exponential increase in exosome research since the 1990s and much data on exosomal cargo have been generated. Databases have been created to provide a compendium of exosomal cargo before it becomes too difficult to manage large data sets. Repeatability of the published data on exosomes is another reason since most labs use variations of different methods developed to isolate exosomes. For example, Vesiclepedia is a manually curated compendium of molecular cargo (lipid, RNA and protein) identified in different classes of extracellular vesicles [64]. Vesiclepedia provides a valuable resource for researchers. The database Vesiclepedia is publicly available and allows users to query and download exosome cargo data based upon user-entered search criteria. EV pedia, Exocarta, ExoRbase and EV miRNA are additional databases that catalogue exosomal proteins and RNA [64]. The databases are driven by community annotations. The data curated in these databases are obtained by isolating exosomes using a variety of different separation techniques that are not included in the database. Exosomal cargo content thus may differ based upon the isolation method used, giving rise to a degree of ambiguity in reported results. To improve experimental reproducibility and transparency, an open-source knowledgebase EV-TRACK tool set was developed. EV-TRACK encompasses data of exosome preparation and exosome characterization [65]. A key feature of EV-TRACK is EV-METRIC that presents the reporting of experimental parameters as a percentage of fulfilled components of a list of nine [66]. These components were decided by the EV-TRACK consortium and were deemed necessary for unambiguous data interpretation and reproduction of experimental data. Thus, EV-METRIC was designed to aid researchers and reviewers in providing a comprehensive overview of the data.

## 2. Current Methods for Exosome Isolation

Exosomes present a rich resource for biomarker discovery and for novel therapeutic strategies including regenerative medicine. Paramount to the success of these goals is the isolation of pure exosomes via a reproducible method. Isolation of functionally competent exosomes is particularly critical for potential therapeutic applications. Exosomes have been purified from various biological fluids such as blood, plasma, cerebrospinal fluid, saliva, urine, and milk. Each biological fluid has different physical and chemical characteristics that need to be considered prior to exosome isolation. In some cases, specific precautions may be necessary for exosome isolation. The end application of purified exosomes is another determinant that influences the method of exosome isolation. To date, several exosome isolation methods have been developed and some of them are based upon exosome characteristics such as their size, density, surface charge, and proteins associated with the exosome surface (Figure 3) [67]. Some of the currently employed exosome isolation methods are listed in Table 1. Most of them are not stand-alone methods and require pre- or post-sample processing steps. Salient methods are briefly discussed below.

Differential ultracentrifugation: The ultracentrifugation method has long been considered the ‘gold standard’ for exosome isolation [10,78] and is the most commonly used method for exosome isolation. Ultracentrifugation was first reported by Johnston and colleagues to isolate exosomes from culture medium of reticulocytes [10]. This technique is based upon the use of centrifugal forces to sequentially separate particles based upon their size and density. Larger and dense particles require low centrifugal forces to sediment first [44,84].

Depending on the nature of the sample, first low-speed centrifugation at 300–2000× *g* removes floating cells, dead and apoptotic cells. A higher-speed centrifugation at 10,000× *g* removes large vesicles and cell debris. Exosomes are pelleted down by ultracentrifugation at 100,000× *g*. The efficiency of exosome isolation depends on a number of factors including acceleration (g), the type of rotor employed and its characteristics (k factor, radius of rotation), and viscosity of the starting sample. Exosome sedimentation efficiency decreases with increased viscosity. Therefore, isolation of exosomes from serum/blood requires higher-speed centrifugation steps for a longer duration as compared to exosome isolation from cell culture medium. Depending on the rotor utilized, relatively large sample volumes can be processed. Ultracentrifugation for exosome isolation requires minimal reagents and consumables and has good reproducibility but requires more time and expensive equipment. The potential of pelleting protein aggregates and other non-exosomal particles has prompted the use of gradient ultracentrifugation that allows exosome isolation based on their buoyant density. Sucrose, Iohexol and iodixanol are generally employed for isopycnic centrifugation [85,86]. This method is time consuming, requires the use of a small sample volume as rotary tubes cannot be overloaded, and yields a small amount of pure preparation of exosomes.

Ultrafiltration: This method is based upon the use of hydrophilic porous membranes with a defined pore—or molecular weight cutoffs—size to trap particles of specific size or mass. Samples such as culture medium or biological fluids are sequentially filtered through membranes with decreasing pore size. This allows retention of large particles at each filtration step and the retentate is enriched with specific particle sizes. Exosomes are recovered from the final retentate by ultracentrifugation. This procedure is fast and has gained popularity [69]. Ultrafiltration is associated with some shortcomings due to membrane clogging and exosome trapping, resulting in moderate exosome yield. Sheer stress of filtration may damage exosomes. 

Size-exclusion chromatography: The principle of this method is based upon protein purification by chromatography. The columns are filled with porous polymer beads or resin such as Sephacryl S-1000, Sepharose 2B, and Sepharose CL-2B [71,83,84,85,86,87,88,89]. When the sample is loaded on to the column, it traverses through the resin. Larger particles travel through the column because they are too large to enter the resin and are the first to be eluted (also referred to as flow through). Small particles that enter the porous resin are eluted by passing buffer through the column. This separation is strictly based upon size or molecular mass and thus allows purification of sub-populations of exosomes. Size-exclusion chromatography is a rapid and reproducible method that does not interfere with integrity of exosomes. The only caveat with this method is that a small fraction of high-density lipoprotein cholesterol and proteins co-elute with exosomes [87]. Additional purification by density gradient centrifugation is required to remove protein aggregates and lipids to obtain pure exosome preparation.

Immunoaffinity capture: This method is based upon the use of antibodies against surface proteins of exosomes such as tetraspanin proteins [78,90,91]. Magnetic beads covalently coated with streptavidin are incubated with biotinylated antibodies against surface proteins of exosomes, also referred to as ‘capture antibodies’. Magnetic beads–capture antibody complexes are incubated with the sample. Unbound sample is removed by magnetizing the beads. Washed beads are processed directly for downstream application without eluting exosomes. This is an easy-to-use method and provides pure preparation of sub-populations of exosomes. However, this method may not be used routinely as antibodies are expensive and this method requires prior knowledge of exosome surface proteins and depends upon the availability of antibodies against them. This method has difficulties with elution of intact exosomes. However, once established, it can be employed for applications such as screening tumor exosomes in the plasma. In addition to magnetic beads, several other platforms such as highly porous monolithic silica microtips, plastic plates, cellulose filters, membrane affinity filters, an agarose sorbent, and microfluidic devices are employed for affinity based exosome purification.

Microfluidics device: The physiochemical and biochemical features of exosomes are exploited at the nanoscale. The microfluidics device requires a small sample volume. The sample is directed to flow through micro-sized channels and exosomes are captured using immunoaffinity methods or entrapment by porous structures [67], size filtration, nanowire trapping, lateral displacement, acoustic nanofiltration, and more recently viscoelastic flow sorting [82]. These devices have garnered much interest as reagents required are in microliter quantities, are portable, and the process of exosome isolation is very rapid and efficient. Thus, microfluidic devices have the potential to be developed for rapid clinical screening and diagnosis. The only disadvantage with this method is the low sample capacity.

Flow cytometry, asymmetrical flow field-flow fractionation, gradient ultracentrifugation, and polymer precipitation are some additional methods that have been established for exosome purification [92,93,94]. It is apparent from the short summary of current isolation methods that no particular approach is suitable for all exosome isolation requirements. Standardization of the method to be utilized for exosome isolation is critical. The optimal isolation method will depend upon the nature of the sample, the amount of starting sample volume, and downstream application. An ideal isolation method would preferably be simple, and not require complex or expensive equipment. In a clinical setting, exosome isolation needs to be scalable, cost effective and rapid. 

Exosome purification from brain: Exosomes have great promise as clinical diagnostic markers for diseases and as prognostic biomarkers. Current exosome isolation methods from biological fluids including blood, urine, and cerebrospinal fluid, while not without challenges, are performed quite routinely. However, to fully harness the potential of exosomes, methods for extracting exosomes from solid tissues need to be developed. With reference to neurodegenerative diseases, the primary pathology is often localized to a specific brain region but toxic proteins subsequently spread to other brain regions. Current mechanisms of toxic protein spread to other brain regions are not completely understood. Exosomes are likely candidates as they are involved in intercellular communication. The scenario that exosomal cargo differs in different brain regions is highly likely. Therefore, isolating exosomes from brain region of interest can potentially be very informative. 

One of the earliest reports of exosome isolation from brain tissue came from Perez-Gonzalez and colleagues [95]. They purified exosomes independently from fresh and frozen mouse brain, and frozen human brain tissues. The isolation method entailed initial papain treatment of tissue followed by filtration to discard cells and debris. Subsequently, the homogenates were subjected to a series of low-speed centrifugations followed by a series of high-speed centrifugations. Exosomes were further purified by fractionation on a sucrose gradient. Using this protocol, the authors reported isolation of intact exosomes from brains [95]. A similar method was utilized by Vella and colleagues except that the sample was not subjected to filtration [96]. The methodology employed by this research group resulted in isolation of exosomes that met all the exosome characterization criteria providing an important tool to neuroscientists. An interesting and different approach to exosome isolation was taken by another research group. Arguing that ultracentrifugation pellets down “unwanted soluble proteins and aggregates”, Gallart-Palau and colleagues homogenized fresh mouse brain and frozen human frontal lobe in 0.1M ammonium acetate buffer supplemented with anti-proteases. Pooled supernatants were mixed with four parts of chilled acetone to precipitate out hydrophilic proteins and similar tissue macromolecules leaving hydrophobic exosomes in solution [97]. In our opinion, there are several drawbacks with this method. First, it is unlikely that ammonium acetate can maintain a neutral pH as the pKa of acetic acid is around pH 4.75 and the pKa of ammonium is around pH 9.25. Thus, brain tissue is homogenized using a buffer with unstable pH and this may affect the charge on macromolecules [98]. Second, although acetone may separate hydrophilic proteins and other hydrophilic molecules from supernatants it may alter the phospholipid bilayer as well and third, increase plasma membrane fluidity suggesting random leakage of exosome contents [99].

## 3. Role of Exosomes in the Central Nervous System (CNS)

Intricate interactions between neurons and glia are a prerequisite for normal function of the nervous system (Figure 4) [100]. Communication in the CNS has been described as multimodal and categorized into wiring transmission (one on one transmission) and volume transmission (one to many transmission) [101]. Wired transmission includes neurotransmission via electrical and chemical synapses engaging transfer of small molecules and ions through gap junctions. Volume transmission, on the other hand, is facilitated by paracrine transmission of signals through the interstitial space or the cerebrospinal fluid. In recent years, however, it became apparent that exosomes contribute to intercellular communication in the CNS [102]. Exosomes achieve their role as cell–cell communicators by exerting their effects through a variety of mechanisms. Exosomes can initiate signal transduction cascades via activation of surface receptors or by delivering their cargo of nucleic acids, proteins, lipids, and other small molecules leading to functional consequences in recipient cells. Exosome release and uptake is bidirectional, allowing for coordination of biologic events. Exosomes are released by all cells in the CNS, including microglia, oligodendrocytes, astrocytes and neurons [103]. Release of exosomes is generally controlled by neurotransmitter signaling in brain. Exposure to the excitatory neurotransmitter glutamate stimulates release of exosomes by mature cortical and hippocampal neurons cultured in vitro. Conversely, AMPA receptor or NMDA receptor antagonists reverse the excitatory activity with a reduced exosome output. These data suggest that exosome release is modulated by excitatory synaptic activity. Furthermore, exosome release is facilitated by neuronal depolarization and calcium influx following treatment with the calcium ionophore, ionomycin [49]. Adenosine triphosphate (ATP) released at synapses acts as co-transmitter activating glial purinergic receptors. ATP promotes exosome release by astrocytes and microglial cells. Associating exosome release with neurotransmitter release suggests that these biological nanoparticles may be preferentially localized to brain regions with high neuronal activity possibly due to greater demand of intercellular communication and coordination of cellular activity in these regions [100]. There are several reviews on this topic that provide detailed account of cell–cell communication via exosomes [52,53,104,105,106,107,108]. Table 2 provides a brief summary of role of exosomes in intercellular communication in brain. Below, we briefly describe exosome-mediated intercellular communication among brain cells. 

Neuron-Derived Exosomes: Cultured neurons of mouse and human origin, and neural stem cells release exosomes [92,117,118]. An interesting observation was made by Chivet and colleagues while studying exosomes released by neuroblastoma cells. They observed cell-specific internalization of exosomes. Exosomes from neuroblastoma cells were readily endocytosed by astrocytes and oligodendrocytes but to a much lesser extent by hippocampal neurons. Exosomes released by neurons in response to synaptic activity, on the other hand, were taken up only by neurons suggesting high specificity of exosome-mediated communications among brain cells [119]. Neuron-derived exosomes have been speculated to participate in synaptic plasticity by removing obsolete proteins and RNA. This is achieved by activating phagocytic function of microglia [120]. Potassium-induced depolarization of cultured human neuroblastoma cells releases exosomes enriched in a set of activity-related miRNAs and the synaptic protein MAP1b. The MAP1b protein is abundantly present in axon growth cones and dendritic spines, and is associated with synaptic plasticity [121]. Depolarization stimulates uptake of soma-dendritic exosomes by neighboring synapses. Internalized exosomes facilitate synaptic strength by rapidly translating mRNAs relevant to synaptic activity in the post-synaptic region [121]. Additional support for the role of exosomes in modulating neurotransmission is provided by studies using primary embryonic cortical neurons that release exosomes containing two microRNAs, Let-7c and miR-21. These microRNAs are ligands for the Toll-like receptor 7 (Tlr7). Activation of the Toll-like receptor 7 plays a negative role in axonal and dendritic growth. Incubation of neurons with Let-7c and miR-21 had a negative impact on dendritic growth in normal neurons and this effect was obliterated in neurons from Tlr7^(−/−)^ knockout mice [122]. It is thus reasonable to assume that neuronal exosomes may modulate neurotransmission by regulating neurite growth during neurodevelopment. Synaptogenesis and synapse maintenance are also influenced by exosomes. The proline-rich transmembrane protein, PRR7 is released by neurons in an activity-dependent manner via exosomes. Internalization of PRR7-containing exosomes by neighboring neurons reduces their number of excitatory synapses. Reduction of excitatory synapses is achieved by activating GSk3β, stimulating proteasomal degradation of PSD proteins, and simultaneously inhibiting exosomal secretion of Wnts [123]. Exosomes are thus influential players for neuron–neuron and neuron–glial communications. 

Astrocyte-Derived Exosomes: Astrocytes represent the most abundant glial cell type in the brain. They are critical for maintenance of brain homeostasis and provide vital functions such as neurotrophic support, modulation of synaptic activity, and plasticity [124,125]. Astrocytic processes maintain contact with blood capillaries in the brain contributing to the maintenance of the blood–brain barrier [126]. In addition to performing these vital roles, astrocytes function as immunocompetent cells in the CNS. Depending upon the context, astrocyte activity can exacerbate inflammatory reactions or promote immunosuppression and tissue repair. At sites of CNS tissue damage, reactive astrocytes form scars that effectively serve as functional barriers to inflammatory cells, thereby controlling CNS inflammation [127]. Reactive astrocytes release neurotoxic substances and inflammatory molecules such as cytokines and chemokines. Some of these functions of astrocytes are partly facilitated by releasing exosomes. In addition, accumulating evidence lends credence to the idea that neuroprotective role of astrocytes are mediated via exosomes. Rat astrocytes cultured in vitro release exosomes containing neuroprotective and neurotrophic factors. In particular, astrocytic exosomes were enriched with fibroblast growth factor 2 and vascular endothelial growth factor [128]. Both are angiogenic molecules and mediate proliferation, axonal regeneration, neurogenesis, and synaptic plasticity [129,130]. Stress causes inclusion of glycolytic enzyme aldolase C in astrocytic exosomes [131]. Increased levels of aldolase C not only induce expression of miR-26a-5p in astrocytes but packaging of significantly elevated levels of miR-26a-5p in exosomes. Internalization of astrocytic exosomes with elevated levels of miR-26a-5p reduced MAP2 and GSk3β protein levels in hippocampal neurons, resulting in decreased dendritic complexity [132]. Exposure of primary rat cortical astrocyte cultures to pro-inflammatory cytokines, IL1β and TNFα resulted in release of exosomes enriched with miRNAs that target proteins involved in neurotropin signaling. Specifically, miR-125a-5p and miR-16-5p targeted NTKR3 and its downstream effector Bcl2. Downregulation of Bcl2 in neurons caused a reduction in dendritic growth and complexity [133]. Thus, astrocytes respond to inflammatory challenge by modulating expression of proteins involved in synaptic stability and neuronal excitability. A reduction in neuronal activity may positively affect neuronal survival following an insult. Exosomes released by astrocytes in response to intracerebral injection of IL-1β in mouse enter the peripheral circulation and promote recruitment of peripheral leukocytes. Thus, these data suggest that exosomes bridge communication between brain and periphery [134]. 

Oligodendrocyte-Derived Exosomes: Fast and reliable propagation of action potential along axons depends upon myelination of neurons. Oligodendrocytes ensheath multiple axons and have the capacity to renew myelin sheaths three times in 24 h [135]. Myelin is particularly enriched in lipids but does contain a number of proteins integral to formation of myelin. Major myelin proteins include proteolipid protein (PLP), myelin basic protein (MBP), myelin-associated glycoprotein (MAG), and 2′,3′-cyclic nucleotide 3′-phosphodiesterase (CNP) [136]. Two proteins, PLP and MBP, are distributed within the myelin sheath and are instrumental in the process of compaction that results in the closely apposed multilayered structure of myelin. In addition to their important function in myelin production, oligodendrocytes provide trophic support to axons and hence are essential for axonal integrity [137]. Communication between oligodendrocytes and neurons is critical to achieve this important function. Recent studies show that oligodendrocyte exosomes are internalized by neurons to enable trophic support to axons. Similar to neurons, exosome release from oligodendrocytes is activity dependent. The neurotransmitter glutamate released by active neurons stimulates an increase in intracellular calcium in oligodendroglia via NMDA and AMPA receptors. Elevated cytosolic Ca^2+^ triggers secretion of CN-P and PLP-enriched exosomes by oligodendrocytes. Once released, exosomes are internalized by neurons completing a two-way direct interaction between oligodendrocytes and neurons [109,138]. In vitro studies utilizing cultured neurons further demonstrated that exosomes released by oligodendrocytes increase neuronal survival under conditions of cell stress [109]. In an extension of these studies, Frühbeis and colleagues demonstrated that CNP- and PLP-deficient mice exhibited impaired exosome release from oligodendrocytes [139]. Importantly, oligodendrocytes isolated and cultured from CNP mutant mice or PLP mutant mice showed a reduction in exosome release thus impairing oligodendrocytes-neuron communication. Lack of communication between oligodendrocytes and neurons via exosomes resulted in axonal degeneration and decreased metabolite support to neurons [139]. These observations suggest that oligodendrocyte exosomes make important contributions to maintenance of neuronal homeostasis.

Microglia-Derived Exosomes: Microglial cells are the macrophages of the brain and spinal cord. They account for approximately 10% of glia. Although their number is comparatively low, they have important functions of maintaining homeostasis by monitoring the presence of tissue infection and damage in the CNS. Under homeostatic conditions, microglia cells are maintained in a quiescent state, but they are constantly scanning their environment. When activated, they exert their phagocytic activity and have the ability to release inflammatory molecules such as cytokines, chemokines. Microglia can acquire two phenotypes, the M1 (inflammatory type) and M2 subtypes (pro-regenerative type), depending upon the stimuli and have the ability to switch between these two opposing phenotypes [140], exemplifying their remarkable plasticity. Similar to neurons and other glial cells, microglia release exosomes to communicate with neighboring cells and cells at a distance. The bioactive cargo of microglial exosomes depends upon the phenotype of the releasing microglia, either pro-inflammatory or pro-regenerative. Microglial exosomes contain all enzymes essential for anaerobic glycolysis and lactate production. Therefore, it is proposed that lactate packaged in exosomes could function as a supplementary energy source for neurons during synaptic activity [141]. Microglial exosomes regulate synaptic transmission by promoting ceramide and sphingolipid production in neurons. Enhanced sphingolipid metabolism positively affects excitatory neurotransmission presenting a novel way by which microglia influence synaptic activity [142]. Faced with infection and/or injury, microglial cells quickly engage in a complex inflammatory response and acquire an M1 phenotype. Microglia now with a pro-inflammatory phenotype (M1 phenotype) release exosomes that help with the development of neuroinflammation. Evidence for microglial exosome involvement in neuroinflammation comes from a study conducted with lipopolysaccharide (LPS). Exposure of microglia to LPS, a major component of Gram-negative bacteria, increases release of exosomes enriched with IL-1β, a proinflammatory cytokine and microRNAs such as miR-155 and miR-375. MicroRNA 155 is an important regulatory microRNA in the immune system and its increased levels are detected in inflammatory diseases [143]. In another study, microglial cells were treated with LPS that increased expression of the N-myc downregulated gene 2 (NDRG2) protein levels. Increased NDRG2 protein in turn stimulated microglia to release miR-375 enriched exosomes. Internalization of miR-375 enriched exosomes reduced cell viability of N2A neurons indicating neurotoxic nature of these exosomes [144]. It appears that exposure to LPS alters microRNAs and proteins packaged in exosomes. The confirmatory evidence came from a study involving BV2 cells. The BV2 cells of C57 Black mice are immortalized microglial cells. When BV2 cells were exposed to LPS, they released exosomes rich in pro-inflammatory cytokines IL-6 and TNFα and proteins related to translation and transcription. The proteomic profile of exosomes analyzed by mass spectrometry identified 49 unique proteins present in exosomes from LPS treated BV2 cells as compared to control BV2 cells. It is worth noting here that exosomes from LPS-activated microglia had 58 proteins while exosomes from control BV2 cells had 37 proteins [145]. The forgoing discussion clearly demonstrates that microglial exosomes are important in facilitating neuroprotective and neuroinflammatory functions of microglial in the CNS.

**Table 2 biology-11-00413-t002:** Role of cell-specific exosomes in intercellular communication among brain cells.

Exosome Cell Source	Mode of Exosome Release	Synopsis of Function
Neuron	Calcium influx, glutamate-mediated synaptic activity regulates exosome release [49,115,117]	(1) Transmission between neurons modulates synaptic plasticity including synaptogenesis and synapse maintenance (2) Modulation of neurotransmission (3) Regulate glutamate transporters in astrocyte (4) Exosomes from neural progenitor cells reduce microglial TNFα, IL-1β and COX-2 levels, thereby protecting retinal photoreceptor cells [146]
Astrocyte	ATP released at synapses triggers exosome release Ultrasound induces a 5-fold increase in the exosome release by human astrocytes	(1) Synaptic function, homeostatic and neuroprotective functions (2) Implicated in propagation of pathogenic proteins in neurodegenerative diseases [147] (3) Exosomes containing glutathione transferase M2-2 protect dopaminergic neurons against aminochrome neurotoxicity [148] (4) Attenuate structural alterations and neuronal death in the hippocampus of traumatic brain injury model rats [149] (5) Ultrasound-induced astrocytic exosomes alleviate Aβ neurotoxicity [150] (6) Exosomal apolipoprotein D, angiogenic factors, and miR-190b promote functional integrity and neuronal survival [128,151,152] (7) Systematic spread of the misfolded and aggregated SOD-1 in ALS via exosomes [153]
Oligodendrocytes (Olig)	Glutamate exposure activates NMDA and AMPA receptors resulting in an increase in intracellular calcium that in turn stimulates exosome release [138]	(1) Prevent oxidative stress and starvation in neurons [154], Exosomes with myelin proteins provide trophic support to neurons and axonal integrity (2) Morphological differentiation of oligodendrocytes and myelin formation (autocrine regulation) [155]
Microglia	Release of exosomes is triggered by ATP which is released at synapses as a co-transmitter and activates glial purinergic receptors Binding of serotonin to microglial 5-HT receptors increases intracellular Ca^2+^ levels that in turn stimulate exosome release [112]	(1) Enhance spontaneous and evoked excitatory transmission in neurons [142]; (2) Alter morphology of dendritic spines in hippocampal cultured neurons [156] (3) Depending upon the phenotype of the parent cell, exosomes exert pro-inflammatory effects and contribute to spread of misfolded proteins, or have pro-regenerative functions by promoting re-myelination in myelin lesions by stimulating oligodendrocyte precursor cell migration [100] (4) P2RX7 antagonist treatment reduces accumulation of misfolded tau accumulation and restored cognitive function in P301S mice through microglia exosomes [157] (5) Exosomal miR-151-3p activates p53/p21/CDK1 signaling cascades to protect neurons and promote axonal regrowth in spinal cord injury model [158] (6) ATP induces enrichment of certain proteins in exosomes that influences astrocyte function [159] (7) Transfer inflammatory molecules, such as IL-1β, IL-18, and TNFα, from glia to glia or glia to neuron, resulting in induction of dopaminergic neuron degeneration in PD patients [108]

## 4. Role of Exosomes in Neurodegenerative Diseases

Many neurodegenerative diseases are associated with accumulation of abnormal, misfolded protein leading to progressive neural and glia dysfunction. Generally, neurodegenerative diseases begin with dysfunction in a discrete brain region. Upon release into the extracellular space, misfolded proteins are transferred to healthy cells and begin to induce endogenous counterpart proteins to misfold like a domino effect [160]. This “infective” process leads to amplification of pathology and spread of disease to wider areas in the brain. Clearly, intercellular communication is important in transmission and progression of neurodegenerative diseases. Exosomes released by all cells in the brain have become an integral player in neuro–glia communications. The ability of exosomes to transport and transfer bioactive cargo such as lipids, RNAs and proteins from one cell to another makes them an attractive candidate as mediators of neurodegeneration. Below, we discuss the impact of exosomes in selective neurodegenerative diseases.

Alzheimer’s Disease: Alzheimer’s disease is the most common form of neurodegenerative dementia characterized by a progressive loss of memory and cognitive abilities. Central to the pathology of Alzheimer’s disease is formation of extracellular aggregates of β-amyloid (Aβ) known as amyloid plaques combined with neurofibrillary tangles of tau. The (Aβ) peptides are derived from sequential proteolytic processing of amyloid precursor protein (APP) by β- and γ-secretases. As APP is an intracellular protein, a hypothesis was formulated that the pathological lesions of neurodegenerative disease involves the physical spread of the misfolded protein from neuron to neuron [161]. However, mechanism(s) for transmission of misfolded proteins remained an intriguing question. One of the earliest reports that started to shed light on the possible mechanisms how Aβ is shed into the extracellular space came from the study conducted by Rajendran and Colleagues [162]. While investigating the location of APP cleavage, they observed that β-secretase cleavage of APP occurs in a subset of early endosomes with subsequent trafficking of Aβ peptide to multivesicular bodies. A small fraction of Aβ peptide associated with exosome membrane was secreted into the extracellular space. Exosomes containing amyloid plaques had exosomal marker proteins, flotillin-1 and Alix. Thus, exosome membrane-associated Aβ peptide may represent a novel mechanism that contributes to amyloid plaque formation in the extracellular space [162]. Since this initial observation, full-length APP and many of its metabolites and several members of secretase family of proteases involved in APP processing were detected in exosomes [163]. Aβ can exist in different conformational states that have different properties and intermediate products of fibril formation. Of these, low-molecular-weight Aβ and protofibrils have been suggested to be particularly neurotoxic and act as seeds for protein aggregation [164,165]. Exosomes isolated from postmortem brains of Alzheimer’s disease patients were shown to have increased levels of Aβ oligomers. These exosomes were internalized when incubated with cultured neurons. Most importantly, they were able to spread Aβ oligomers to other neurons, causing cytotoxicity [166]. The initial observations were confirmed by incubating exosomes containing APP with primary cultures of normal neurons in vitro [167] and in vivo [168]. The question is why are Aβ or Aβ oligomers packaged into exosomes? Do neurons sense the toxic nature of Aβ or Aβ oligomers and hence try to clear toxic proteins from intracellularly present Aβ or Aβ oligomers similar to transferrin receptor in reticulocytes [7,8]? Monoubiquitination is required for sorting into MVB/exosomes. That raises a second question as to whether Aβ undergoes ubiquitination to be sorted into MVB/exosomes. The amyloid precursor protein (APP) has five lysine residues (Lys-724, Lys-725, Lys-726, Lys-751, and Lys-763) at its C-terminal end [169]. These residues have been mutated individually or in combination to examine effect on APP processing to form Aβ peptide. Ubiquitination of APP at Lys-726 mediated by the F-box and leucine-rich repeat protein2 (FBL2), a component of the E3 ubiquitin ligase, reduced Aβ generation [170]. On the other hand, APP ubiquitination at Lys-763 sequestered APP in the Golgi complex and prevented APP maturation [171]. Inhibition of ubiquitination by substitution of all five lysine residues to arginine in C-terminal fragment of APP (C99) prevented efficient degradation of APP and accumulation of protein in structures with Golgi-like appearance. This was attributed to a deficiency in endoplasmic reticulum-associated degradation. The C99 undergoes cleavage by γ-secretase to produce Aβ [172]. Mutation of three lysine residues (Lys-724, Lys-725, and Lys-726) simultaneously caused the protein to be retained in the limiting membrane of endosomes instead of becoming internalized into intraluminal vesicles of MVBs [173]. When all five lysine residues were mutated to prevent ubiquitination, the protein did not efficiently sort to MVB/exosomes and a selective increase in Aβ40 was observed [174]. This finding is comparable to the presence of Aβ40 in amyloid deposits in cerebral amyloid angiopathy [175]. While it is evident that ubiquitination of APP may direct the protein to MVB/exosomes, direct evidence using neuronal cultures or in vivo models is required to prove that APP, Aβ, or Aβ oligomers are monoubiquitinated for their targeting to MVB/exosomes and that they are not ubiquitinated to be targeted for endoplasmic reticulum-associated degradation (ERAD). 

Another pathological hallmark of Alzheimer’s disease is abnormally phosphorylated tau protein in neurofibrillary tangles (NFT). Tau is a cytoplasmic protein known to stabilize microtubules. Increasing evidence suggests that the pathological tau protein can spread between cells, recruiting native tau to form aggregates. More recent data implicated exosomes as a carrier of tau protein [176,177]. Cell scrutiny suggested that exosomes involved in tau pathology originated from microglial cells. Exosomes from microglia transferred tau protein to neurons. To obtain experimental evidence, studies were carried out in which either exosome biogenesis was inhibited or microglial cells were depleted of tau. Results from these studies showed attenuation of tau deposition in normal neurons [178]. The authors suggested that microglia phagocytose tau-containing cytopathic neurons and recycle tau through exosomes thus incriminating exosomes in propagation of Alzheimer’s disease. In healthy brains, a number of protein kinases and phosphatases are responsible for phosphorylation and de-phosphorylation of tau, respectively. Dysregulation of these important enzymes may lead to abnormal phosphorylation pattern of tau in AD. A recent study compared exosomal protein cargo by Nano-LC–MS/MS. Exosomes were purified from human-induced pluripotent stem cell (iPSC) neurons expressing the AD familial A246E mutant form of presenilin 1 (mPS1) and normal human iPSC neurons. A total of 1117 proteins were identified in both exosome groups and 733 proteins were common to both populations of exosomes. Among differentially associated proteins with mPS1 were phosphatases and protein kinases and their protein levels associated with mPS1 exosomes was significantly lower as compared to control neurons [179]. Furthermore, these exosomes contained distinct proteins absent in control exosomes. Specifically, the distinct proteins were those involved in extracellular matrix structure and function suggesting another mechanism for propagation of tau pathology in AD [179].

The major emphasis of research in Alzheimer’s disease had been on neurons. However, studies revealed that atrophy of astroglia occurs at early stages of the neurodegenerative process. The lack of neuronal support from atrophied astroglia results in disruptions of synaptic connectivity, loss of synapses, and, imbalance of neurotransmitter homoeostasis. At a later stage of Alzheimer’s disease, astrocytes and microglia become activated and release inflammatory molecules and neurotoxic substances. Neurotoxic chemicals result in neuroinflammation and neuronal death leading to atrophy of brain [180]. In early stages of Alzheimer’s disease, microglial cells activated through Toll-like receptor 4 acquire a neuroprotective role and clear Aβ [181]. Phagocytosis and degradation of purified polymorphous beta-amyloid protein deposits and Aβ associated with exosomes were confirmed using cultured microglial cells [182,183]. Curiously, astrocytes appear to relieve microglial cells of their neuroprotective function. Meda and colleagues noticed that incubation with Aβ peptide activated glial cells and subsequently resulted in inflammation [184]. Co-culturing with astrocytes or culturing in astrocyte-conditioned medium-inhibited phagocytic action of microglia. Inhibition of microglial phagocytosis was highly specific as the conditioned medium from fibroblasts had no effect on microglial phagocytic activity. From these studies, it is evident that astrocytes released signals in the form of soluble factors that interfered with phagocytic activity of microglia [182]. Roles of astrocytes and microglia can be reversed following chronic stimulation of microglia. 

Several independent studies lead to the conclusion that exosomes released by neurons, astrocytes, and microglia act as scavengers and soak up seed-free soluble Aβ to promote Aβ aggregation that is internalized by microglia for degradation [185,186,187]. This observation is not surprising given the closely monitored intercellular communication among brain cells (briefly discussed in Section 3). As mentioned earlier, the Aβ aggregates interact with glycosphingolipids, ceramide, and/or the GPI-anchored protein PrPc (cellular prion protein) present on the surface of exosomes and on neurons [185,187]. Neuronal exosomes bind Aβ aggregates more efficiently as compared to astrocytic or microglial exosomes due to abundantly present ganglioside GM1 and sialylated glycosphingolipids especially trisialoganglioside GT1 on their surface [188,189,190]. Astrocytic exosomes are enriched with the sphingolipid ceramide [187]. The Aβ aggregates bound to astrocytic exosomes are internalized by neurons and are directed to mitochondria, causing mitochondrial clustering and simultaneously increasing the fission protein Drp-1 levels. At the outer membrane of mitochondria, exosomal Aβ forms a complex with the ADP/ATP transporter, voltage-dependent anion channel 1 and activates caspases. Active caspases induce neurite fragmentation and eventually neuronal cell death [191].

To better understand possible dynamic interactions between proteins, lipids and RNA in Alzheimer’s disease, high-throughput techniques have recently been applied. Cohn and colleagues took an ‘omics’ integrative approach to analyze microglial exosomes [192]. In this study, the authors isolated microglial exosomes from the parietal cortex of late-stage Alzheimer’s disease patients. They performed an integrative analysis combining proteomic, transcriptomic, and lipidomic analyses. They used shotgun proteomics, which refers to a bottom-up protein analysis where proteins are characterized by analysis of peptides released from the protein through proteolysis [193], targeted lipidomics, and NanoString nCounter technology a multiplex nucleic acid hybridization technology [194]. Using this combinatorial approach, the research group showed a significant reduction in homeostatic microglia markers P2RY12 and TMEM119, and increased levels of disease-associated microglia markers FTH1 and TREM2. In addition, tau protein levels in AD brain-derived microglial exosomes were significantly higher suggesting that microglia-derived exosomes appear to be important in the spread of tau pathology. Synaptic and neuron-specific proteins were also differentially enriched in AD brain-derived microglial exosomes. The authors hypothesized, however, that the synaptic and myelin-specific proteins have been phagocytosed prior to entering microglial exosomes. The lipidomic analyses revealed a pro-inflammatory phenotype and a potential defect in acyl-chain remodeling. Lastly, miRNAs associated with immune and cellular senescence signaling pathways were increased in AD brain-derived microglial exosomes [192]. These data suggest that a significant change in the molecular composition of exosomes reflects changes in microglia consistent with a diseased state.

Clearly, we have amassed a large amount of data that implicate exosomes in Alzheimer’s disease. An important question is to what extent do exosomes promote or prevent clearance of misfolded proteins? Additionally, elucidating the contributions made by exosomes released from different brain cells will further promote our understanding of disease transmission.

Parkinson’s Disease: Parkinson’s disease is one of the most common age-related brain disorders—it is primarily considered a movement disorder, with typical symptoms of a resting tremor, rigidity, bradykinesia and motor instability [195]. Additionally associated with this disease are cognitive decline, depression, and psychosis [196]. Pathologically, it is characterized by degeneration of nigrostriatal dopaminergic neurons and the presence of Lewy bodies which contain misfolded α-synuclein protein in surviving neurons. Alpha synuclein is detected in many body fluids including cerebrospinal fluid and plasma [197,198]. Alpha synuclein is found in culture medium when cells expressing α-synuclein are cultured in vitro [199]. Considering that α-synuclein has been detected extracellularly in the absence of an extracellular sorting signal, how does this protein reach the extracellular space? Several investigators have examined the mechanism of α-synuclein secretion including the role of exosomes in α-synuclein secretion and pathology. 

A first indication that exosomes could indeed be involved in the pathogenesis came from an in vitro study employing SH-SY5Y cells expressing α-synuclein. The authors demonstrated extracellular secretion of α-synuclein via exosomes in a calcium-dependent manner and suggested their involvement in spread of Parkinson’s disease pathology [200]. Following this study, α-synuclein-containing exosomes have been identified from different cells, cerebrospinal fluid, and plasma of Parkinson’s disease patients [201,202,203]. However, there is a lot of variability reported in different studies [204]. Interestingly, the amount of α-synuclein in exosomes is relatively low as compared to free α-synuclein detected in cerebrospinal fluid or conditioned media. A close examination revealed that α-synuclein-expressing neuroglioma cells use two pathways to release α-synuclein oligomers, one through exosomes and a second, direct release of free α-synuclein as oligomers [205]. Although exosomes have low levels of α-synuclein, exosomes are more effective in exerting their toxic effects than the free α-synuclein. Alpha-synuclein oligomers present in exosomes were internalized more efficiently by human H4 neuroglioma cells as compared to free α-synuclein oligomers [205]. As a follow up of this study, this group of researchers reported similar results using exosomes purified from cerebrospinal fluid of Parkinson’s disease patients. Efficient uptake of exosomal oligomerized α-synuclein was observed in human H4 neuroglioma cell cultures as compared to free α-synuclein oligomers [206]. 

Evidence for secretion of free α-synuclein to the extracellular space or fluid came from a study examining the role of vacuolar protein sorting 4 (VPS4) in loading α-synuclein to multivesicular bodies. Normally, the VPS4 regulates sorting of proteins to multivesicular bodies. In Parkinson’s disease, VPS4 directs α-synuclein to lysosomes for its degradation and to recycling endosomes for extracellular secretion of α-synuclein [207]. When lysosomal function was inhibited, release of α-synuclein packaged in exosomes was seen to increase in cultured SH-SY5Y cells [208]. The endosomal fraction of α-synuclein escapes degradation in conditions of lysosomal impairment [209]. 

What facilitates incorporation of α-synuclein in exosomes? Alpha synuclein is found to be associated with lysosomes [210] and endosomes extracted from mouse brains [207], suggesting that α-synuclein may be recruited to early or late endosomes. Late endosomes may fuse with lysosomes for degradation of α-synuclein or result in the formation of multivesicular bodies to release α-synuclein associated with exosomes. Evidence suggesting that the endosomal pathway may promote incorporation of α-synuclein into exosomes comes from ubiquitination studies. Davies and colleagues found that α-synuclein is ubiquitinated by the E3 ligase Nedd4 and ubiquitinated α-synuclein is targeted to endosomes [211]. This process is negatively regulated by USP8 [212]. SUMOylation appears to be another mechanism for sorting of α-synuclein to exosomes. SUMO protein modification is a ubiquitin-independent ESCRT mechanism that appears to regulate α-synuclein release via exosomes [213]. Neutral sphingomyelinase-2 hydrolyses sphingomyelins to ceramide and phosphocholine. Inhibiting neutral sphingomyelinase-2 with cambinol (DDL-112) for five weeks reduced α-synuclein aggregates and exosome biogenesis and improved motor function in PD mouse model [214]. 

A fundamental question to understand the pathogenesis of Parkinson’s disease is how do exosomes relay the toxic effects of α-synuclein? Exosomes may aid Parkinson’s disease pathogenesis by promoting aggregation of α-synuclein due to their lipid and/or protein composition thus facilitating uptake of α-synuclein by cells. Several studies pointed out that exosomes contain α-synuclein as oligomers. Several cellular processes and enrichment of certain molecules within cells cause α-synuclein oligomerization and often in combination with other proteins. Exosomal ganglioside lipids GM1 or GM3 accelerate α-synuclein aggregation [215]. A combination of ceramides and neurodegeneration-linked proteins including α-synuclein and tau in exosomes is capable of inducing aggregation of wild-type α-synuclein [216]. Oxidation of two adjacent amino acids, methionine [Met(38)] and tyrosine [Tyr(39)], results in aggregation of γ-synuclein and seed aggregation of α-synuclein. Neuronal exosomes containing γ-synuclein upon internalization can cause aggregation of intracellular proteins in astrocytes, resulting in synucleinopathies [217]. Levels of Golgi complex localized the gamma adaptin ear-containing, ARF-binding protein 3 (GGA3) were downregulated in postmortem substantia nigra of PD patients as compared to controls. GGA3 induces oligomerization of α-synuclein in endosomes, resulting in secretion of α-synuclein oligomers [218]. In another study, researchers reported interaction between α-synuclein and the autophagy protein, LC3B that resulted in formation of detergent-insoluble oligomeric aggregates. Alpha-synuclein oligomers are deposited on the surface of late endosomes and are eventually secreted out of human pluripotent stem cells through exosomes [219]. 

Another intriguing question is how toxic α-synuclein is transferred between cells. Several studies have addressed transfer of pathological α-synuclein from neuron to neuron and neuron to microglia and vice versa. The first evidence for transmission of α-synuclein in Parkinson’s disease pathogenesis came from two independent research groups. In one study, neurons from the substantia nigra were transplanted into the striatum of an individual with Parkinson’s disease. When examined fourteen years later, transplanted neurons were positive for α-synuclein aggregates similar to host dopamine neurons in the substantia nigra of the subject [220]. Simultaneously, Li and colleagues reported similar findings using two human subjects with Parkinson’s disease [221]. These results demonstrate that cell to cell transmission is a continuous, insidious process. Once exosomes were established as an important entity in intercellular communication, exosomes packaged with α-synuclein were identified. Neuron to neuron transmission occurs through internalization of α-synuclein-containing exosomes in Parkinson’s disease pathology [208]. Normal embryonic dopaminergic neurons transplanted into striatum of rat brain overexpressing human α-synuclein quickly endocytosed α-synuclein and was found in early endosomes. Results of this study lent strong support to data obtained in 2008 which showed transmission of disease in vivo [220,221,222]. Lipid peroxidation is associated with late onset of Parkinson’s disease. Lipid peroxidation results in the formation of 4-hydroxynonenal. Exposure of primary neurons to this product increases aggregation of endogenous α-synuclein. Extracellular vesicles released by primary neurons contained cytotoxic oligomeric α-synuclein. Endocytosis of these extracellular vesicles caused degeneration of healthy neurons in vitro. Injection of neuronal extracellular vesicles containing cytotoxic oligomeric α-synuclein to striatum of normal healthy mouse resulted in transmission of α-synuclein pathology not only in striatum but surrounding brain regions as well [223]. Implantation of exosomes from bone marrow mesenchymal stem cells rescued the pathogenic features of Parkinson’s disease by altering the inflammatory microenvironment in the substantia nigra and repairing the injury to dopaminergic neuron nerves These exosomes were enriched with Wnt5a [224]. The long-term rescue efforts of exosomes are not clear from the study. 

Microglia are a double-edge sword in the CNS as they can be either neuroprotective or neurotoxic. Incubation of microglial cell line BV2 with α-synuclein released an increased number of exosomes enriched with MHC class II molecules and membrane TNFα. Internalization of these exosomes by neurons was neurotoxic suggesting a role for microglia in α-synuclein-induced neurodegeneration [225]. The question is how α-synuclein is internalized by microglia. Microglial cells selectively express Toll-like receptor 2 (TLR2) that acts as a ligand of α-synuclein. Binding of α-synuclein to TLR2 activates microglia. Since α-synuclein is present on the surface of exosomes, they are internalized by microglia. The excessive exosome uptake by microglia causes an inflammatory response [226], inhibition of autophagy, and reduced scavenger activity. Reduced phagocytosis of α-synuclein-containing exosomes was seen in mouse microglia and human monocytes from aged donors. This observation suggests an age-dependent predisposition to incidence of misfolded proteins in Parkinson’s disease [227], which is in line with the onset of Parkinson’s disease occurring predominantly after 60 years of age. 

Surprisingly, only limited studies have addressed the role of astrocytes in pathogenesis of Parkinson’s even though these cells are known to play important roles in the CNS. As mentioned earlier in this section, astrocytes internalize neuronal exosomes containing γ-synuclein. In this case, γ-synuclein induced aggregation of intracellular proteins of astrocytes [217], thereby interfering with astrocyte function. Involvement of astrocytes in Parkinson’s disease came from another study involving mutations in the leucine-rich repeat kinase 2 (LRRK2) protein. The LRRK2 protein plays a role in vesicle trafficking, possibly via phosphorylation of Rab GTPase substrates [228]. Mutations in LRRK2 gene are associated with late-onset Parkinson’s disease. In particular, the G2019S mutation resulted in increased kinase activity of LRRK2 by auto-phosphorylating Ser residue at 1292 [229]. Ser(P)-1292 LRRK2 protein was detected in exosomes from urine of Parkinson’s disease patients [229,230]. Astrocytes generated from LRRK2 G2019S patient-derived-induced pluripotent stem cells (iPSC) released exosomes. These exosomes had an abnormal shape and were enriched with LRRK2 and phospho-S129 α-synuclein. Dopaminergic neurons internalized LRRK2 G2019S astrocytic exosomes. Interestingly, internalized exosomes accumulated to a greater extent in neurites as compared to soma suggesting that neurite localized astrocytic exosomes either interfered with neuronal function or were processed for recycling through neurites [231]. The neurotoxic effects of α-synuclein are caused by an increase in Ca^2+^_i_ through activation of voltage-operated Ca^2+^ channels and a significantly increase in mitochondrial Ca^2+^ sequestration [232]. 

A number of studies examined microRNAs packaged in exosomes from cerebrospinal fluid and plasma of Parkinson’s disease patients, cultured cells, and rat models of Parkinson’s disease [233,234,235,236]. Several microRNAs were differentially associated with exosomes from the diseased samples as compared to controls. Pathway analyses of differentially present microRNAs in exosomes suggested their involvement in the following pathways: ubiquitin-mediated proteolysis, long-term potentiation, axon guidance, cholinergic synapse, gap junction, dopaminergic synapse, and glutamatergic synapse. One microRNA, miR-23b-3p, was strikingly reduced in Parkinson’s disease exosomes. MicroRNA-23b-3p binds to 3′-untranslated region of α-synuclein. Reduction of miR-23b-3p in exosomes leads to upregulation of α-synuclein mRNA [235], thereby increasing expression of α-synuclein protein in Parkinson’s disease. The underlying cause of Parkinson’s disease is not only an increase in α-synuclein but a combination of several dysregulated pathways that lead to etiology of the disease. Continuous dysregulation of these pathways is perhaps important to the progression of the disease. Can we re-set dysregulation of affected pathways? At least one study examined this possibility. Exosomes from adipose-derived stem cells (ADSC) are enriched with miR-188-3p. This microRNA targets NAcht leucine-rich repeat protein 3 (NLRP3) and cell division protein kinase 5 (CDK5), and both targets are involved in autophagy. Internalization of ADSC exosomes reduced autophagy in MN9D cells [237]. 

In summary, there has been substantial interest in exosome research in the context of Parkinson’s disease. From the foregoing discussion, it is apparent that exosomes are important mediators of α-synuclein transmission among brain cells. In addition, the ability of exosomes to transfer proteins and miRNAs contributes to pathogenesis.

Amyotrophic Lateral Sclerosis: Amyotrophic lateral sclerosis (ALS) is a late-onset, fatal neurodegenerative disease with a median survival of only 2–5 years—it affects upper motor neurons which project from the cortex to brain stem and spinal cord, as well as lower motor neurons that project from the spinal cord to muscles. Patients develop progressive muscle paralysis and death usually occurs due to respiratory failure. Most cases are sporadic but some are familial cases. ALS is characterized by misfolding of Cu/Zn dismutase (SOD-1) [238] and TAR DNA-binding protein 43 (TDP-43) [239]. SOD 1 is a cytosolic mitochondrial enzyme involved in clearance of superoxide molecule, while TDP-43 is a highly conserved nuclear RNA/DNA-binding protein involved in RNA processing. Post-translational modifications such as cleavage, hyper-phosphorylation and ubiquitination of TDP-43 can lead to cytoplasmic accumulation and aggregation of TDP-43. Both SOD-1 and TDP-43 are packaged in exosomes [240,241]. By overexpressing both wild-type and mutated SOD-1 in NSC-34 motor neuron-like cells, Grad and colleagues observed that misfolded SOD-1 protein was transferred from cell to cell via exosomes in addition to direct uptake of SOD-1 protein aggregates by micropinocytosis [241]. Studies have suggested that astrocytes may play a role in pathogenesis of ALS. Exosomes released by primary astrocyte cultures expressing mutant SOD-1 efficiently transferred mutant SOD-1 protein to spinal neurons, causing selective motor neuron death [242]. A study utilizing a SOD-1 transgenic mouse model demonstrated that mutant SOD-1 was enriched in exosomes derived from both neurons and astrocytes, suggesting that these two cell types may contribute to spread of pathology in ALS [153]. TDP-43, another protein involved in pathogenesis of ALS, was detected in exosomes purified from cerebrospinal fluid of ALS patients [243], supporting the idea that exosomes contribute to disease propagation. Indeed, cerebrospinal fluid enriched with TDP-43-containing exosomes was able to promote accumulation of toxic TDP-43 in human glioma U251 cells [244]. Furthermore, TDP-43 oligomers present in exosomes were transmitted intercellularly [245]. Interestingly, levels of exosomal TDP-43 (full-length protein and C-terminal fragments) are upregulated in brains of ALS patients. When Neuro2a cells were exposed to exosomes from ALS brains, TDP-43 was redistributed in the cytoplasm of Neuro2a cells [246]. Compared to other neurodegenerative diseases, research into the pathogenesis of this devastating fatal disease is much more limited. Much of the research has been performed in vitro. With refinement of exosome isolation techniques from brain tissue it is hoped that we will have a clearer picture of the role-played by exosomes in spread of ALS.

From the foregoing discussion on neurodegenerative diseases, it is clear that exosomes provide a vehicle for transmission of misfolded proteins (or toxic proteins) thus playing a role in the propagation of disease. Certainly, transmission of toxic proteins through exosomes is not the only mode of transmission. However, an important aspect to consider is that exosomes may provide a suitable environment for proteins to aggregate and stay in an aggregated form. At this time, we do not understand whether exosomes lie at the core of pathology of neurodegenerative diseases or whether they are released as a consequence of the disease process. A better understanding of how toxic proteins are packaged into exosomes and how they are transferred to naive cells will provide important insight into pathogenesis of devastating diseases involving misfolded proteins. This information will provide opportunities for improved therapeutic strategies and hopefully personalized treatments. 

Exosomes and the blood–brain barrier: The blood–brain barrier is a physical barrier between brain and the peripheral circulation, controlling a strict influx and efflux of molecules to maintain the homeostasis. Accumulating evidence suggests that exosomes have the remarkable ability to cross the blood–brain barrier from both directions. Exosomes carry cargos of membrane and cytosolic proteins and genetic material such as mRNAs, non-coding RNAs including miRNAs that otherwise generally do not cross plasma membrane. Exosomes released from cancer cells have been shown to destroy the blood–brain barrier through action of microRNA-181c, leading to actin mislocalization and perhaps, resulting in breakdown of blood–brain barrier integrity. Such leakiness of the blood–brain barrier is also seen in cases of neurodegeneration often as a result of neuroinflammation. Furthermore, glioblastoma-specific mRNAs have been detected in exosomes in the peripheral circulation [247]. Experimental evidence suggests that exosomes can cross the blood–brain barrier from the periphery and localize in the brain. Analyses of fluorescent or luciferase-labeled exosomes demonstrated that they have the ability to accumulate in the brain from the periphery [248,249]. Exosomes loaded with siRNA were able to deliver their cargo to neurons, microglia and oligodendrocytes in brain when administered intravenously [208]. Exosomes derived from hematopoietic cells can be transferred to Purkinje cells in the brain and importantly were able to modulate gene expression in these cells. This observation suggests that transfer of exosomes via the blood–brain barrier can have functional implications. The ability of exosomes to cross the blood–brain barrier presents a great potential for exosomes as a drug delivery system. Equally important is that uptake of the exosomal cargo by recipient cells can have profound functional impacts on the CNS. Thus, understanding how exosomes traverse the blood–brain barrier bidirectionally can have great therapeutic potential and diagnostic utility.

Concluding remarks: As the exosome field is witnessing exponential growth, it is perhaps an understatement to say that there is a requirement for more uniformity in exosome isolation and characterization methods. Refinement of exosome isolation in an in vivo setting will definitely enable the discovery of novel biological functions of exosomes. Many exosome studies have been performed using cells cultured in vitro. Future studies involving animal and clinical research will be a key to unlocking the potential of exosome biology. Particularly, a better understanding of the role played by exosomes in pathogenesis of neurodegeneration will pave the way for new therapeutic avenues. This is specifically significant as the aging population increases and with it a growing incidence of neurodegenerative diseases. The biological content of exosomes can be harnessed for biomarker discovery aiding in diagnosis and prognostic follow up studies. This is of particular importance as exosomes are present in most biological fluids and the biological cargo is stable and protected within the boundaries of exosome membranes. 

Future perspective of exosomes in neurodegeneration: Slow and progressive deterioration of the quality of life of patients suffering from neurodegenerative diseases has a devastating effect not only on patients but also on family members and medical professionals. Researchers worldwide are engaged in efforts to identify biomarkers that will unequivocally detect early signs of these crippling diseases. Olfactory dysfunction resulting in loss (anosmia) or reduction (hyposmia) of smell is considered as an early sign of the neurodegenerative diseases [250,251,252]. Unfortunately, impairment of smell is not unique to neurodegenerative diseases alone as exposure to drugs of abuse such as alcohol, viral infections such as COVID-19, trauma, or simple sinusitis or polyposis nasi also interfere with olfactory abilities [253,254,255]. Identification of gene-specific mutations; post-translationally modified and/or misfolded protein levels in CSF; and PET imaging have made significant contributions to our understanding of disease progression. More recently, exosomes show great promise to help us understand the pathogenesis of disease spread and in identifying exosome-associated unique protein(s), non-coding RNA(s), lipid(s), or metabolite(s) as biomarker(s) for a specific neurodegenerative disease. Biomarker discovery for neurodegenerative diseases is particularly critical because these diseases progress silently sometimes for decades before obvious clinical manifestations. Substantial neuronal death has already occurred in late stages of the disease when diagnosis is made. Hence, current treatments are only palliative once disease is diagnosed in late stages of the disease. Identification of changes that take place before the appearance of visible signs of disease is therefore crucial to our ability to identify biomarkers of disease. Identification of biomarkers in the neurodegenerative disease field is currently impeded due to lack of systematic analysis of exosomes from beginning of the disease. Since biological fluids are enriched with exosomes, analysis of exosomes on a regular basis from members of families with known mutations for neurodegenerative diseases and disease models may be one opportunity to identify biomarkers. However, such studies require commitment of funding agencies, family members, and researchers since these are long-term studies and come with a substantial price tag. 

## Figures and Tables

**Figure 1 biology-11-00413-f001:**
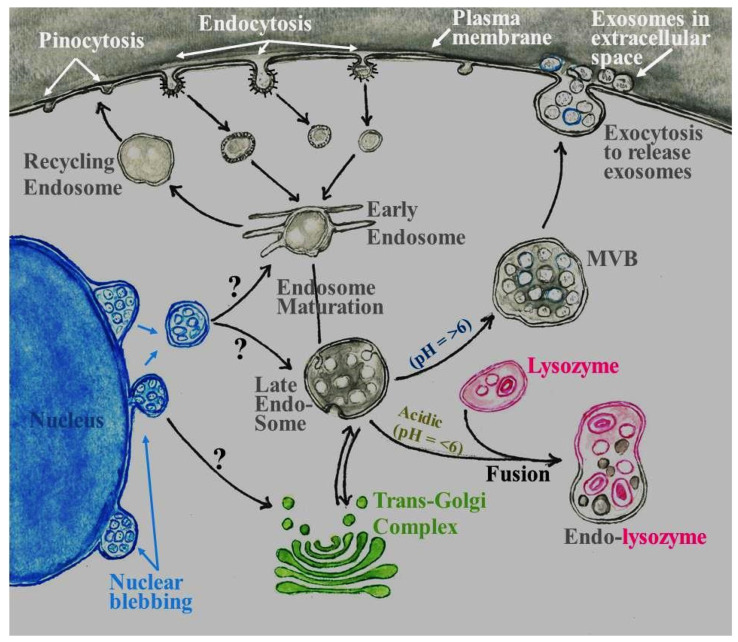
Cellular events related to biogenesis and release of exosomes from the cell. Pinocytosis, clathrin (bristle-coated)-mediated and clathrin-independent endocytosis phosphate) from the *trans*-Golgi network and phagosomes (not shown here). The lumen of late endosomes becomes acidic due to the progressive concentration of ATP-driven H+ pumps in the vesicle membrane. Acidification is a complex phenomenon not completely understood. If the pH in the lumen is below 6, then late endosomes fuse with lysosomes for degradation and recycling of their content. However, if the pH in the lumen of late endosome is above 6, then they escape fusion with lysosomes and subsequent hydrolysis. During maturation of early endosomes to late endosomes, intraluminal vesicles arise from inward invaginations of the internal membrane. RNAs, proteins, and lipids are incorporated into developing intraluminal vesicles through ESCRT-dependent and ESCRT-independent mechanisms. Late endosomes packed with intraluminal vesicles have a multivesicular appearance and are identified as multivesicular bodies. They migrate towards and fuse with the plasma membrane to release their intraluminal vesicles, i.e., exosomes into the extracellular space. Nuclear blebbing appears to generate structures similar to multivesicular bodies (here referred to as MVB type II) that eventually detach from the nuclear membrane and become cytosolic. The MVB type II packed with RNA and/or genomic DNA may receive RNA-binding proteins from the nucleus and/or the *trans*-Golgi network and fuse with early/late endosomes to export their intraluminal vesicles into the extracellular space. Figure 1 was drawn by authors based upon [20,24,25,26,27,28].

**Figure 2 biology-11-00413-f002:**
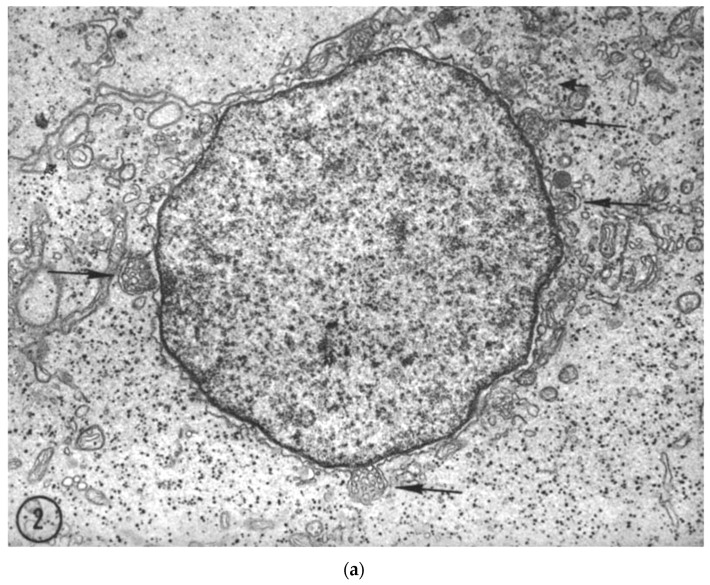
(**a**) Section through a cell of gas gland of perch fish showing nuclear blebbing (long arrows). Nuclear blebs are filled with “fine vesiculae” with a diameter of 51–72 nm and 168–290 nm. Ultrastructure appearance of nuclear bleb is comparable to multivesicular bodies that originate from endosomes in the cytoplasm. Encircled 2 is the original label from Kilarski and Jasiński manuscript. Magnification: ×5250 [24]. (**b**) Illustration depicts the development of round-shaped intraluminal vesicles (A-C1) and tubular-shaped vesicles (A-C2) from single in-foldings of the internal membrane of nuclear bleb (multivesicular body type II). The mechanism of vesicle formation in nuclear blebs is similar to development of ‘intraluminal vesicle’ in cytoplasmic endosomes. (**c**) Illustration labeled as encircled 9 depicts formation of a connecting neck that separates from the external nuclear membrane, tearing the nuclear bleb from the nucleus, and setting it free in the cytoplasm. Nuclear blebs are filled with small vesicles. The intermediate vesicle (a small round structure) formed from the neck of nuclear bleb is seen between nuclear membrane and separated nuclear bleb. Figure 2 is adapted with permission from Wincenty Kilarski, Andrzej Jasiński; The formation of multivesicular bodies from the nuclear envelope. J. Cell Biol. 1 May 1970; 45 (2): 205–211. doi:10.1083/jcb.45.2.205 [24].

**Figure 3 biology-11-00413-f003:**
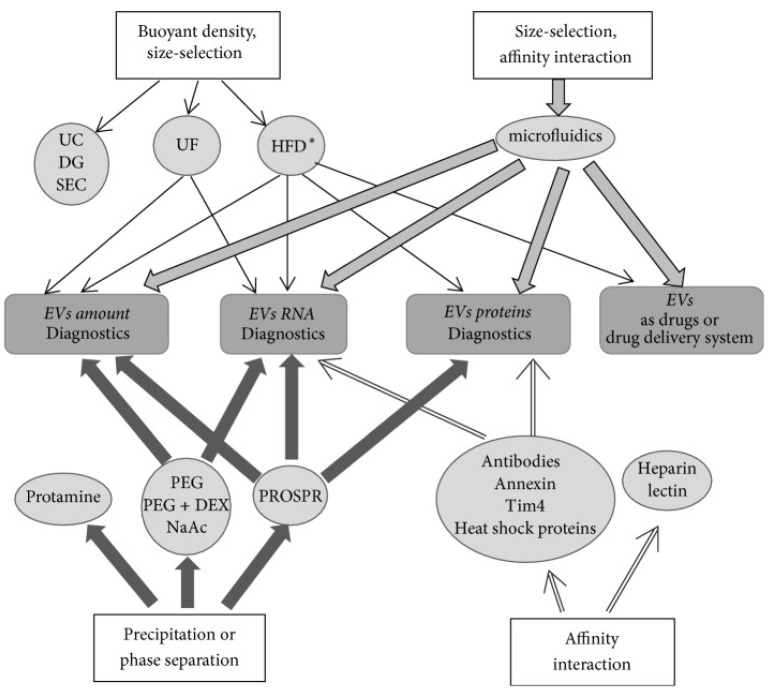
Commonly employed methods for exosome isolation and their downstream applications. Several methods developed to date for exosome isolation are based upon their buoyant density, Scheme 1. 13 and 1.19 g/mL, the ultracentrifugation method (UC) was further refined to isolate pure preparations of exosomes based upon their buoyant density by isopycnic ultracentrifugation over sucrose or iodixanol density gradient (DG). To remove high protein content in proteinurias, the ultracentrifugation method was combined with size-exclusion chromatography (SEC). Time constraints associated with the ultracentrifugation method led to development of the filtration method (UF). Passing dilute samples such as culture medium and urine through a nano- or micro-membrane concentrator allowed exosome purification. Membrane clogging and poor exosomal protein recovery made this method unpopular for downstream applications such as RNA isolation and high-throughput applications. Another method developed to isolate exosomes from very dilute biological fluids is hydrostatic filtration dialysis (HFD *) in combination with centrifugation. In the HFD method, the sample is dialyzed through a dialysis membrane with a cutoff of 1000 kDa molecular weight under hydrostatic pressure, allowing unwanted macromolecules to be removed from the sample and exosomes are recovered by centrifugation. The exosome precipitation method based upon solubility or dispersibility and aggregation was developed to process several samples simultaneously. Commercial kits for exosome purification are based upon this strategy. In this method, urine or ascites are mixed with polyethylene glycol (PEG) 6000 or in combination with protamine (plus PEG 35,000) and incubated either overnight or 1 h on ice. T resulting turbid mixture is centrifuged to recover exosomes that are used for downstream RNA and protein analyses. Other reagents used for exosomes precipitation are (1) 0.1M sodium acetate pH 4.75 (salting out method) (NaAc), (2) a 4-fold volume of cold acetone (−20 °C) (PRotein Organic Solvent Precipitation and abbreviated as PROSPR), (3) and a mixture of PEG and dextran solution (up to 1.5%) (a two-phase method that allows separation of proteins from exosomes). Identification of exosomal surface proteins led to development of immunoaffinity capture-based methods using antibodies to exosome surface proteins such as Tim4, annexin, EpCAM, common exosome surface protein, CD63 and heat shock proteins (affinity interaction). Affinity for lectin (that binds to glycoproteins on the surface of exosomes) and heparin (that binds to heparan sulfate proteoglycans on exosome surface) was exploited to develop exosome isolation methods. A number of platforms such as magnetic beads, highly porous monolithic silica micro-tips, surface of plastic plates, cellulose filters, membrane affinity filters, an agarose sorbent, and microfluidic devices were used for affinity based exosome purification methods and they have different merits for their use in diagnostic, prognostic, as drug delivery systems, and for exosomal cargo detection and analyses. *Adapted with permission from* Konoshenko MY, Lekchnov EA, Vlassov AV, Laktionov PP. Isolation of extracellular vesicles: General methodologies and latest trends. Biomed Res Int. 2018 Jan 30; 2018:8545347. doi:10.1155/2018/8545347. PMID: 29662902; PMCID: PMC5831698; Copyright © 2022 Maria Yu. Konoshenko et al. This is an open access article distributed under the Creative Commons Attribution License, which permits unrestricted use, distribution, and reproduction in any medium, provided the original work is properly cited [67].

**Figure 4 biology-11-00413-f004:**
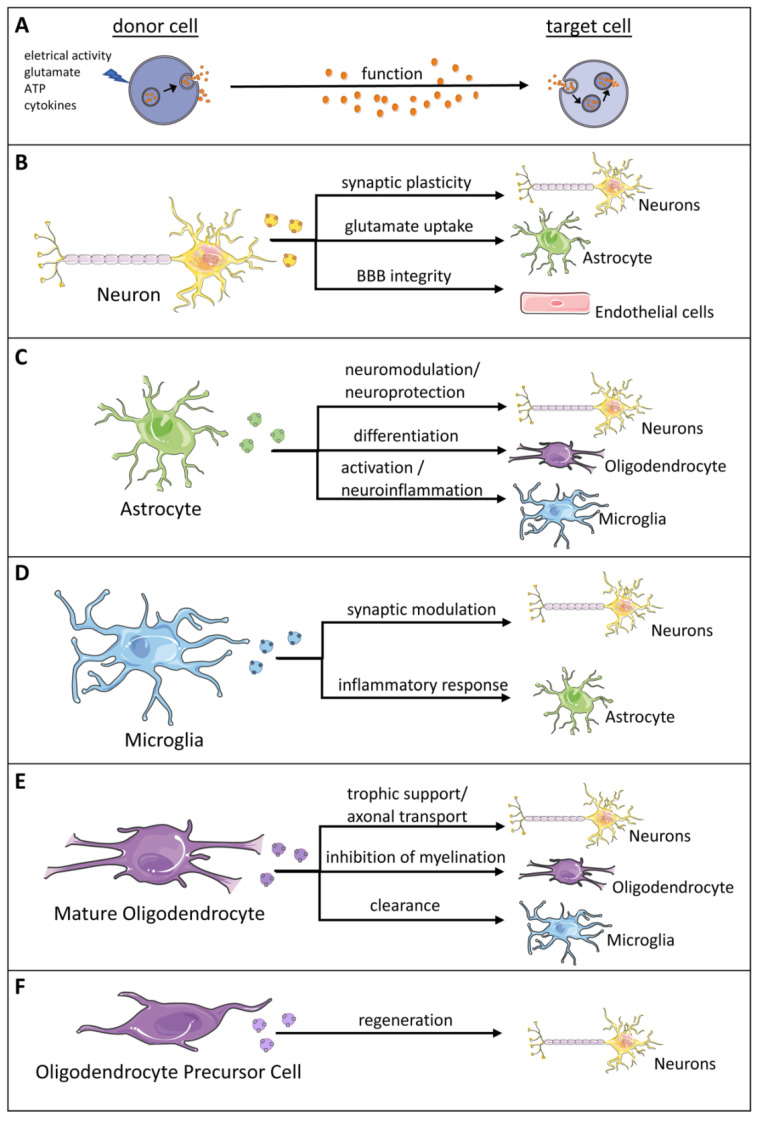
Exosome-mediated intercellular communication among brain cells. Exosomes from a parent cell may have several target (or recipient) cells within brain to ensure maintenance of homeostasis. (**A**) Exosome release in the CNS occurs in response to neurotransmitter signaling and therefore is controlled by electrical activity. For example, exposure to potassium chloride depolarizes neurons, often known as electrically active neurons. Depolarized neurons release glutamate that binds to and stimulates AMPA and NMDA receptors on oligodendrocytes. Activation of glutamate receptors increases intracellular calcium levels in oligodendrocytes. An increase in intracellular calcium induces release of exosomes by oligodendrocytes [109]. ATP-mediated stimulation of P2 purinoceptor subtype, P2X7 stimulates exosome release by microglia [110] and astrocytes [111]. Serotonin also stimulates exosomes release by microglia [112]. (**B**) Neurons regulate certain functions of neurons, astrocytes, and endothelial cells through release of exosomes [113]. In particular, neuronal exosomes eliminate synapses and stimulate phagocytosis by microglial cells [114]. Neuronal exosomes released in response to synaptic activation, control expression of glutamate transporters in astrocyte to regulate extracellular glutamate levels [115]. (**C**) Astrocyte-derived exosomes modulate neuronal function and exert a neuroprotective role. Astrocyte exosomes activate microglia and are an important player in differentiation of oligodendrocytes. (**D**) Microglial exosomes modulate synaptic function of neurons and communicate with astrocytes to induce neuroinflammation. Microglial exosomes can mediate both neuroprotective and neurodegenerative roles associated with microglia. (**E**) Mature oligodendrocyte exosomes promote axonal transport and help maintain a clean microenvironment through communication via microglia and astrocytes. (**F**) Oligodendrocyte precursor cell-derived exosomes assist in neuroregeneration [116]. *Adapted with permission from* Schnatz, A., Müller, C., Brahmer, A. and Krämer-Albers, E.-M. (2021), Extracellular Vesicles in neural cell interaction and CNS homeostasis. FASEB BioAdvances, 3: 577–592 [100]. doi:10.1096/fba.2021-00035; Copyright © 2022 The Authors. FASEB BioAdvances published by the Federation of American Societies for Experimental Biology. This is an open access article under the terms of the http://creativecommons.org/licenses/by-nc/4.0/License (3 March 2022), which permits use, distribution, and reproduction in any medium, provided the original work is properly cited and is not used for commercial purposes.

**Table 1 biology-11-00413-t001:** Various methods developed and employed for exosome purification.

Principle of the Method	Method	Methodology/Procedure	Application
Size and density based	Differential centrifugation (ultracentrifugation method)	Involves successive centrifugation steps with an increase in centrifugation forces and durations to separate small particles from large particles; reliable method but cumbersome [10]	Biological fluids, cell culture medium, RNA-seq and proteomic analyses
Size and density based	Isopycnic or gradient centrifugation using: - Sucrose - Iodixanol - Optiprep - Sucrose and deuterium oxide cushion	Involves density based separation of exosomes; exosomes have a density between 1.13 and 1.19 g mL^−1^; pure preparation but low yield [67,68]	Exosome preparation with relatively higher purity and can be used for exosomal RNAs and proteins studies; clinical-grade purified exosomes
Size based	Microfiltration Ultrafiltration	Involves sequential filtration through “low protein” binding membranes with decreasing pore size, thereby excluding particles bigger than exosomes [69]; ultrafiltration involves the use of amicon filters [70]; membrane clogging and inability to exclude small particles from exosomes	Low-density biofluids such as urine and culture medium
Size based	Size-exclusion chromatography using porous polymer beads or resin such as Sephacryl S-1000, Sepharose 2B, and Sepharose CL-2B	Involves loading of samples on to the resin packed column and sequential elution of particles as they traverse through the resin. Larger particles are eluted as flow through and small particles are eluted by passing buffer through the column [71]. Rapid and reproducible method, does not affect exosome integrity. A small fraction of high-density lipoprotein cholesterol and proteins co-elute with exosomes	Best for dilute biofluids, tissue culture medium, and tissue exosomes
Size based	Hydrostatic filtration dialysis	Involves dialyzing out molecules through dialysis membrane with 1000 kDa cutoff size and recovery of exosomes by centrifugation [72]	Best for dilute biofluids and long-term storage of biofluids for exosome studies at a later date
Precipitation or aggregation based	PEG 6000	Involves precipitation of exosomes by addition of PEG 6000 (8–9% final concentration) to the sample; variable amounts of contaminants such as proteins, protein complexes, lipoproteins, and nucleoproteins in exosome preparations [73]	Allows exosome isolation from dilute samples and processing of several samples simultaneously
Precipitation based	PEG 35,000 plus protamine	Involves precipitation of exosomes using positively charged protamine (0.25 mg/mL) in the presence of PEG 35,000 [74]	Best for plasma, saliva, and culture medium. For exosomal RNA analysis
Precipitation based	Sodium acetate	Involves addition of acetate to 0.1M pH ~4.75 to neutralize surface charge and salt-out exosomes; depends on both pH and salt concentration, rapid method [75]; may affect surface properties of exosomes	Best for culture medium or dilute biological samples; may have applied value for quick exosome isolation
Precipitation based	Organic liquid (PROSPR)	Involves precipitation of plasma proteins by addition of cold acetone (−20 °C) four times the volume of plasma; separation of precipitated proteins and recovery of exosomes require additional methods such as ultrafiltration or ultracentrifugation [76]; acetone may coagulate membrane proteins and dissolve exosome membrane lipids	Best for small volume samples such as plasma
Precipitation based	PEG + dextran	Involves repeated extraction using two-phase system consisting of 4.5% PEG 25,000~45,000 and 1.5% dextran (450,000 to ~650,000 molecular weight); removes >95% of the serum proteins [77]; difficult to recover exosomes from dextran	Best for plasma and culture medium; used for RNA isolation
Affinity based	Antibodies such as Tim4, annexin, EpCAM, CD63, and heat shock proteins conjugated to paramagnetic beads, porous monolithic silica microtips, plastic plates, cellulose filters, membrane affinity filters, agarose beads, and microfluidic devices	Involves sample incubation with a surface (e.g., paramagnetic beads) conjugated to antibody [78,79], exosome elution by incubation with IgG and exosome concentration by ultracentrifugation; highly specific but difficult to elute exosomes [79]; purifies a sub-population of exosomes that can be beneficial when trying to detect exosomes from specific parent cells in biological fluids	Isolation of exosomes with specific exosomal markers, e.g., cancer-specific proteins; larger sample volumes can be processed
Affinity based	Lectin Heparin	Involves incubation of samples with lectin that binds to glycoproteins on exosome surface or heparin that binds to heparan sulfate proteoglycans on exosome surface [80,81]	Rapid, allows isolation of exosome for RNA analysis; may be valuable for medical clinical diagnosis
Physiochemical and biochemical such as hydrodynamic and dielectrophoretic properties based	Microfluidic devices - Immunoaffinity - Size filtration - Lateral displacement - Acoustic nanofiltration - Nanowire trapping - Viscoelastic flow sorting	Involves flow of sample through a small device; exosome purification depends upon their hydrodynamic, dielectrophoretic, and biochemical properties [82,83]	Rapid, microscale isolation for medical clinical diagnosis

## Data Availability

Not applicable.

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
