# Peer review of "Small but Mighty—Exosomes, Novel Intercellular Messengers in Neurodegeneration"

_biology, 2022, doi:10.3390/biology11030413_

Round 1

Reviewer 1 Report

The Review article entitled, “Small but mighty- exosomes, novel intercellular messengers" by Kumari et al., discusses exosomes their origin, purification and their role in central nervous system. Exosomes are released by all cells in our body and their cargo consisting of lipids, proteins and nucleic acids. They summarized many methods for exosome isolation and a summary for exosomes released by different cells in the brain and their role in maintaining CNS homeostasis is also discussed. Authors particularly focused on progression of Alzheimer’s, Parkinson’s disease, and amyotrophic lateral sclerosis.

Altogether this is an important and timely review article, this reviewer has certain suggestions that would help produce a more comprehensive overview of the topic:

Comments:

1, Figure 1 and 2A quality may be improved (high resolution) if possible, remove dark background.

2, The English of manuscript can be polished (minor) check typo errors to throughout the manuscript.

3, The title of the manuscript is “Small but mighty- exosomes, novel intercellular messengers” but authors mainly focused on exosomes released by different cells in the brain. Hence this reviewer wants to change the title according the theme of that review.

4, Remove numbers from Keywords.

5, Authors can include a table describing role of exosomes in the central nervous system/ neurodegenerative diseases.

6, Authors should summarize current methods for exosome isolation in a table.

7, A separate paragraph is needed for future prospective .

Author Response

We thanks the reviewer for their time and insight. We have made appropriate changes to the manuscript and all changes are highlighted in red. We have included additional references as additional information was required for updating the manuscript as per reviewer's suggestions.

Sincerely,

Meena Kumari

Reviewer 2 Report

The review covers interesting topics and would be worth improving.

The main problem is that historical information is interspersed with recent findings. The reader he is required to continually verify if what he is reading refers to something recent or to very old.  

The authors should dedicate the first long paragraph for the historical analysis and evolution of the discoveries relating to exosomes, as in fact it has been done, and proceed with the evaluation of the most recent discoveries for the remainder of the review.

For example lanes 171-173“some miRNAs enriched in exosomes are not detected in the parent cells suggesting a selective RNA sorting /packaging system during biogenesis of  exosomes [41,43]” 2007, 2008,2012 respectively. New works on miRNA and nucleic acid exosomes content comes out every day,

it would be interesting to report the most important ones.

Lanes 604-605 “amyloid-β and protofibrils have been suggested to be particularly  neurotoxic and act as seeds for protein aggregation [115,116]”  dated 2004 and 2010 respectively. About the role of amyloid-β , new mechanisms of action and new important hypothesis  are being recently reported and it is dispersive to be informed of what was known 10-20 years ago.

-The final part of paragraph 1 where the author talk about the content of nucleic acids would be very interesting, but is a bit confused and it would be better to distinguish even with speculative hypotheses what the transport of the various types of RNA and genomic DNA may represent

The hypothesis  presented  regarding differences between nuclear or cytoplasmatic origin of EV should be explained more clearly

-Figure 3 Given that several paragraphs are dedicated to the description of the purification methods, Figure 3 should give some more details,  for example on the role of the indicated proteins without having to consult the work of Konoshenko et al

-Figure4 “(A) Exosome release in the CNS occurs in response to neurotransmitter signaling and therefore is controlled by electrical activity”. - The general mechanism of information transfer between the various nerve cells by exosomes is described Figure 4 A? the authors should explain this in the legend.

The title reports the general concept of “novel intercellular messengers” it is not appropriate to describe the subject of the review that reports abundant historical reconstruction, then I propose to insert in the title the sentence “intercellular messengers in in neurodegeneration”.

Author Response

We thank the reviewer for his/her time to review our manuscript and providing feedback. We have addressed all the concerns and incorporated all the changes in the revised manuscript. All changes are highlighted as red text.

Sincerely,

Meena Kumari

Round 2

Reviewer 2 Report

The manuscript has been adequately revised